# God Spots in the Brain: Nine Categories of Unasked, Unanswered Questions

**W. R. Klemm**

College of Veterinary Medicine & Biomedical Sciences, Texas A&M University, College Station, TX 77843, USA; wklemm@cvm.tamu.edu

**Abstract:** Neurotheology is an emerging academic discipline that examines mind-brain relationships in terms of the inter-relatedness of neuroscience, spirituality, and religion. Neurotheology originated from brain-scan studies that revealed specific correlations between certain religious thoughts and localized activated brain areas known as "God Spots." This relatively young scholarly discipline lacks clear consensus on its definition, ideology, purpose, or prospects for future research. Of special interest is the consideration of the next steps using brain scans to develop this field of research. This review proposes nine categories of future research that could build on the foundation laid by the prior discoveries of God Spots. Specifically, this analysis identifies some sparsely addressed issues that could be usefully explored with new kinds of brain-scan studies: neural network operations, the cognitive neuroscience of prayer, biology of belief, measures of religiosity, role of the self, learning and memory, religious and secular cognitive commonalities, static and functional anatomy, and recruitment of neural processing circuitry. God Spot research is poised to move beyond observation to robust hypothesis generation and testing.

**Keywords:** God Spots; neurotheology; brain scans; review

---

## 1. Introduction

Neurotheology is an emerging academic discipline that examines mind-brain relationships in terms of the inter-relatedness of neuroscience, spirituality, and religion. Key intersections in various early stages of investigation include a potential evolutionary basis of religion, psychology, mental health, and brain pathology; and myths, rituals, and mystical experiences (Newberg 2018).

The field of neurotheology emerged from metabolic brain-scan discoveries, made in a few pioneering laboratories, showing that specific areas of the brain become more metabolically active when people have religious experiences (d'Aquili and Newberg 1999; Newberg et al. 2002; McNamara 2009; Newberg 2010, 2018). Scholars have responded to these findings in various ways, ranging from intrigue, to indifference, and to dismissive labeling of these areas as "God Spots" in the brain.

If neurotheology is to escape the skepticism, investigators in this discipline must find ways to get beyond the limitations of merely identifying God Spots. Neurotheology has not yet consolidated its definition, ideology, purpose, and scholarly and applied strategies for development of the budding discipline. Resistance to acceptance by the scholarly community is compounded by the perceived inherent conflicts between science and religion.

Neurotheology faces three categories of obstacles: (1) many scholars in each partnering discipline are hostile to the other discipline, (2) it is unclear what each partner brings to the marriage of disciplines, and (3) both disciplines have quite different histories and experiences with spawning scholarly hybrids (such as neuroscience's birth of neuropsychology, neuro-education, neurophilosophy, etc.) and religion's birth of multiple subdisciplines (such as sacramental theology, practical theology, philosophical theology, etc.). An open question is whether neurotheology can thrive as a discipline

if confined to identification of God Spots (Klemm 2019a). An alternative, more expansive view of neurotheology would include a worldview promoting the incorporation of relevant aspects of neuroscience and mental health into religious teaching, counseling, mission, doctrine, and practice (Klemm 2019b).

We must consider if God-Spot research is reaching a point of diminishing returns, nearing a dead end, where one must ask: "Now what?" The key issue, as yet unexplored, is what to make of brain area activity that links to specific religious thought. When areas X, Y, and Z "light up" during prayer, for example, what does that tell us about cognitive mechanisms in general or about theology in particular? Is it possible to use brain-scan technology to augment our understanding of cognition in general? Can this technology be used to understand more completely certain kinds of religiosity? Can God Spot findings help us interpret the now-abundant scan data obtained in secular contexts? Can we use God Spot activity to help us become better thinkers or more dedicated to our religious beliefs and practices?

Central concern needs directing toward testing the selectivity of religious and spiritual cognition in God Spots, which undoubtedly engage in other non-religious kinds of neural processing. Perhaps brain-scan studies of God Spots can unmask underlying non-religious cognitive processes. There are important unknowns in neuroscience, particularly regarding emotions (Johnston and Olson 2015) and cognition (Sternberg 1999), and both are both central to religious belief and practice.

We should also recognize the intractable limitation of metabolic brain scans, which can only be a proxy for the actual underlying electrical signaling that occurs during thought. Even so, the scan data's identification of God Spots opens new doors for cognitive neuroscience.

The identification of God Spots leaves us uncertain whether there are any useful next steps. God Spots are real, and they must mean something. They might even be put to practical use. Creative thinking and clever experimental designs will be necessary for any of the potential to be achieved.

## 2. Categories of Issues that Can Be Explored by Brain Scans

### 2.1. Avoiding the New Phrenology

Well before metabolic brain scans identified God Spots, scans had identified numerous localized regions of the brain that seemed to have dedicated cognitive functions. We have known for decades about localized function in the somatosensory and motor cortices, but new research goes far beyond that to implicate functional neural architecture in specific higher order cognitive processes (Kanwisher 2010). There are, for example, cortical zones associated with directed attention, executive control, face recognition, body and body-part recognition, inhibition of movement responses, place location, theory-of-mind processes, color recognition, visually guiding reaching and grasping, visually presented word-form analysis, and other specific cognitive functions.

Such a view of functional specialization has led to a proliferation of diagrams in research papers and textbooks showing such "hot spots" and implying a new kind of phrenology. Until about 1840, phrenology was a popular pseudoscience based on the now-discredited notion that measurement of bumps on the skull indicated mental traits.

Yet, even the very brain-scan research that argues for functional specialization also provides evidence that the concept is incomplete and flawed. First, it seems clear that a given thought typically involves more than one "hot spot" of activity, often distantly separated even in the opposite hemisphere. Processing of a given thought likely activates a network of neurons, with one or a few zones serving as a network hub.

Brains have billions of neurons linked in a global workspace network that contains innumerable sub-networks. Neural networks are built from groups of interconnected neurons forming a local electrical circuit, one or more of which may serve as regulatory nodes for activity within that circuit and the interactions of that circuit with other connected circuits. For example, in a simple circuit involving only a few neurons, the presence of an inhibitory neuron acts as a node to exert major regulation of the

entire circuit. Interacting circuits have different degrees of spatial separation, which in turn influence the temporal dynamics of communication among connected circuits. Microcircuits involving just a few neurons may serve as nodes in a larger circuit complex.

Imaging studies have identified a default network that processes spontaneous and self-generated thought, including mind wandering, mental simulation, social cognition, autobiographical retrieval, and episodic future thinking consisting of a set of linked cortical areas, particularly the precuneus/cuneus, midline and posterior inferior parietal lobule, and posterior cingulate cortex. The executive network directs cognitive processes that require externally directed attention, including working memory, relational integration, and task-set switching, most prominently includes the dorsolateral prefrontal cortex (DLPFC) and the anterior cingulate cortex (ACC) (Heinonen et al. 2016). There is also a salience network that involves the dorsal anterior cingulate (dACC) and orbital frontoinsular cortices with robust connectivity to subcortical and limbic structures (Seeley et al. 2007). There also seems to be a creativity network that couples with the default network during divergent thinking and domain-specific artistic performance (Kenett and Faust 2019). Depending on task requirements, the default and control networks may interfere or cooperate with each other (Beaty et al. 2016). A connectivity analysis study revealed that during divergent thinking, all these networks become more coupled (Beaty et al. 2016). The authors concluded that "In general, we contend that the default network influences the generation of candidate ideas, but that the control network can constrain and direct this process to meet task-specific goals via top-down monitoring and executive control." A connectivity analysis study that compared high- and low-creative people revealed that their connectivity patterns were vastly different (Kenett and Faust 2019).

If thoughts, secular or religious, are processed in specific networks, we should then shift our research in the direction of functional network analysis. Specifically, a series of questions arise. What regulates how the brain allocates network resources for the processing of a given kind of thought? How does the brain assess how much processing resource is needed for a given cognitive task? How does the brain identify which portion of the global network is available at any given instant for a given task? What is the mechanism of such allocation? How does the brain determine when to release an engaged sub-network for operation on another processing task? Does a specific sub-network shift to a different locus after extensive learning or damage to its original locus?

Neuroscientists and theologians should have their own reasons to be interested in God Spot research that compares analogous religious and secular thought in the context of network dynamics. Belief, hope, empathy, gratitude, anxiety, reason, willed action, learning, and psychological actualization are some of the kinds of thought that are manifest in both religious and secular thought. Neuroscientists can use God Spot research to catalog and understand analogous religious and secular kinds of thought and learn how and where neural resources are allocated for processing. One testable prediction is that there is nothing particularly unique about God Spot content in terms of neurophysiology and that God Spot research might guide study of secular cognition.

For theologians, a network perspective opens the door for exploring the inter-relationships between different kinds of theological thinking, for example, the degree of relationships and overlap among different kinds of prayer or worship and the ways they might enrich each other. Theologians might come to realize that religious teaching and counseling could be more effective when they capitalize on familiar secular ways of thinking. We might hypothesize that different kinds of prayer generated in different parts of the global network reflect underlying secular thought processes that can be enlisted to augment a religious application.

## 2.2. Cognitive Neuroscience of Prayer

A religious believer might contend that the appearance of God Spots in the brain indicates that in some sense God has entered our brain and mind. Prayer has been the usual focus of God Spot research because it is a common way believers choose to interact with God. Also, prayer is one spiritual activity people can perform without moving (brain scans require cessation of movement to eliminate artifacts).

Another problem with prior brain-scan research on prayer is that it fails to disambiguate the act of prayer from its content. As an example of this point, a study by Newberg et al. (2003) evaluated brain scans of nuns chanting a phrase during "meditative prayer," but there was no way to control for the various possible cognitive content in such prayer.

Various investigators have compared ritual and spontaneous prayers with secular activities. For example, Azari et al. (2001) acquired PET scans while six religious and six nonreligious participants recited either the first verse of Psalm 23, a nursery rhyme, or a set of phone-card instructions. Reciting the Bible verse was associated with increased activity in the right dorsolateral prefrontal cortex in the religious, but not the non-religious subjects. Religious participants also showed more activation in zones responsible for bodily alertness (in the dorsomedial frontal cortex) than did non-religious people.

Another study, conducted with the now-popular fMRI type of scanning, found that the activation of God Spots depended on whether the prayer was scripted or improvised (Schjødt et al. 2009). Scans were obtained from participants as they recited the Lord's Prayer, recited a well-known rhyme, made wishes to Santa, counted backward from 100, or engaged in the non-scripted act of improvising a personal prayer. During improvised prayer, brain activity increased in the areas of the brain associated with the ability to take another person's perspective (e.g., theory-of-mind regions in the temporo-parietal junction, temporopolar region, left medial prefrontal cortex). This pattern was less robust during recitation of the Lord's Prayer. Increased activity occurred in the caudate nucleus activity during recitation of the Lord's Prayer, but less activity occurred there during improvised prayer. The precuneus (thought to process self-referential thought) was active during both improvised prayer and wishing to Santa.

Neubauer (2014) employed fMRI in people praying, expressing love or gratitude to a loved one, or imagining and naming animals. In the prayer and love conditions, the medial prefrontal cortex and posterior cingulate showed elevations above baseline, suggesting theory-of-mind connections. Prayer linked to more robust emotional arousal than did the love condition. In an fMRI study by Beauregard and Paquette (2006), brain scans taken of nuns while they prayed to achieve "a state of union with God" revealed activation in the right medial orbitofrontal cortex, right middle temporal cortex, right inferior and superior parietal lobules, right caudate, left medial prefrontal cortex, left anterior cingulate cortex, left inferior parietal lobule, left insula, left caudate, and left brainstem.

Activation of the striatum was noted by Schjødt et al. (2008), who observed dorsal striatal (right caudate head) in Danish Christians when they improvised individualized prayers that they found to be rewarding.

These studies by no means exhaust the range of possibly useful areas of future prayer research. Currently published studies have not thoroughly investigated the difference in God Spots depending on the cognitive nature of prayer involved. Not all forms of prayer are the same. The published literature has not systematically compared activity when the same person participates in each of the common kinds of prayer, such as adoration, petition, confession, or thankfulness. These involve distinctive thought and may therefore selectively recruit different neural resources.

*2.3. Biology of Belief*

Brain-scan researchers believe that fMRIs yield insight into the biology of belief, both religious and secular (Harris et al. 2008, 2009; Kapogiannis et al. 2009a, 2009b; Beauregard and Paquette 2006). God Spot research raises the question: "When brain areas X, Y, Z become active during a religious experience, has some neural process opened the gates to a neural network that allow God to enter within us?" Or conversely, does God Spot activation provide us with a gateway to think about God in various ways? However, these are probably not questions that neuroscience is equipped to answer.

Past religious experiences may have programmed certain networks of the brain to process a specific set of religious thoughts. Initial exposure to religious doctrines or sets of ideas may function as a mental stimulus. That stimulus elicits representations that need to be processed somewhere in the brain.

What about the opposite possibility of programming disbelief? Can we program our brain to close the gates to God in our neural circuitry? Debates rage over whether belief or disbelief is a voluntary decision-making process. It is not clear what God Spot research can contribute to the debate over whether a person can choose to believe or disbelieve. However, secular brain-scan research has addressed decision-making processes (Hollmann et al. 2011). Perhaps such scanning could be relevant for studying religious belief formation. Of particular relevance might be scans of "regions of interest" (ROIs) identified in secular decision-making research under conditions involving formation of religious beliefs. For example, does activity in these ROIs differ in before and after "born again" experiences? Are there ways to understand how religious ideas come to be believed or disbelieved? Are there ways of "packaging" religious ideas in ways that are more likely to be admitted to positive neural processing and decision-making? Brain scans might be a way to compare such packaging for effectiveness.

### 2.4. Measures of Religiosity

One approach using the God Spot frame of reference would be to ask Christians about their confidence in the belief that Jesus is the Savior, or ask Buddhists about the degree of confidence that they can reach a state of Nirvana. The ratings might show correlation with the extent and intensity of God Spot activity.

Many people may be deceiving themselves or others when claiming their status as a believer or non-believer. Some may even intentionally lie about it. The idea that brain scans can unmask deception has been studied, and it is not clear that brain scans are reliable for lie detection (Stix 2008). Regardless of the veracity of what people claim, brain scans can measure the intensity of an individuals' thought or experience, in terms of the magnitude of increased activity and the number of active brain areas. Perhaps God Spot research could reveal whether neural activity conforms to what people think they believe. For example, people may think they are praying fervently, but the brain scan could indicate a lack of focus and distraction by other thoughts.

### 2.5. Role of the Self

The neural basis for sense of self is only vaguely understood (Klemm 2012). In terms of embodiment, we do know that the somatosensory and motor cortices are dedicated to the processing of this kind of selfhood information. This part of the cortex, known as the homunculus, maps the body parts. However, sense of self has other dimensions, such as abstract identity of selfhood, distinction of objects and other beings, location in time and space, ownership of thought and emotion, and self-agency. I have suggested that the homuncular system might engage in the dimensions of selfhood that go beyond embodiment (Klemm 2020). Surely, the neural network modules that process various dimensions of selfhood must at least access and interact with the homunculus or a stored memory of it. There are many neurophysiological correlates of the sense of self, both electrophysiological and from brain scans. However, these correlates have not been specifically linked to the various dimensions of self-hood (Klemm 2011b). More to the point, we must test to see how these correlates change when we get "outside ourselves" in religious and spiritual thought and practice. Religious thought and practice can have intense engagement of the self with a simulated perception of God. Does this get magnified in the brain areas processing the engagement? And would the effect vary with the aspect of selfhood that is involved?

### 2.6. Learning and Memory

Not much is known about the possible changes in God Spot activity with repetition and sustained experience with religious ideas and practice. Brains are changed by programming, as is well-documented in education, psychotherapy, and the accumulated experience of daily living and age. For secular programming, the systematic self-correcting learning process known as "deliberate practice" is the most effective way to program new behavior into one's brain. Classic examples include learning to play the piano and perfecting a golf swing. The same principles used in these classic

examples should apply to religious thought and practice. Repeating brain-scan identification and activity intensity of relevant God Spots in a given person for a specific religious thought or feeling can indicate how well the religious programming is progressing. Moreover, if God Spot activity was used in biofeedback, religious programming might be faster and more reliable. For example, such an approach might be used in biofeedback training that would help believers pray more intensely or help Buddhists reach their sense of Nirvana.

Multiple brain-scan studies conducted in non-religious contexts show that the number of brain regions used to process a given thought can decrease dramatically as a process is learned through continued repetition (Raichle et al. 1994). This is because as learning progresses, fewer neural resources are needed. For example, with repeated practice of a verbal learning and a motor task, activity shifts from left frontal, anterior cingulate, and right cerebellar hemisphere to activity in the Sylvian-insular cortex. In this study, similar changes were also observed in the second task in a very different domain, namely when navigating a maze. Some areas of the brain (right premotor and parietal cortex and left cerebellar hemisphere) became significantly less active as learning developed with practice (Petersen et al. 1998).

Another example comes from a study of ballet dancers in training, who were monitored at four time points over 34 weeks in which they visualized the movements they were learning. Initial learning and performance at seven weeks led to increased activation in cortical regions during visualization of the dance being learned when compared to the first week. However, at 34 weeks, scans showed reduced activation compared to week seven (Bar and DeSouza 2016).

Unlike computers, where the circuitry used in response to programming remains the same no matter how often a process is repeated, brains can release portions of circuitry for other functions as the neural network masters the required information processing. Do the number of God Spots decrease with repeated spiritual thought? What is the brain-scan effect of repeated spiritual activity, such as praying the Catholic rosary or Muslim "call to prayer?" Can brain scans of religious experiences reveal a similar reduction of neural resource allocation as a given religious thought or experience is repeated? If so, would the religious process offer any insight on how this conservation of neural resources is mediated? Can the "brain-training" effect of repeated religious experiences produce lasting improvements in neural resource allocation for secular thought? Are the training effects of religious thoughts and experiences any different from repeated secular thought and experience?

Assuming that repeated religious thought experience produces neural resource conservation, would pairing a religious thought or experience with related secular information affect the memory formation of the secular information? For example, would memorizing a sequence of historical events become more robust if one repeatedly prayed about remembering these events, as opposed to just repeatedly rehearsing the events without the religious context?

### 2.7. Religious and Secular Cognitive Commonalities

We can catalog different kinds of thought. For example, Gain (2018) identified some 21 cognitive categories for emotional processing. Secular and religious thoughts can have certain features in common. Therefore, we could construct a table that shows specific kinds of religious thoughts and their secular counterpart (for a few examples, see Table 1). There are semantic difficulties with such a scheme, but common understanding of linguistic meaning allows us to make approximate labeling of various kinds of thought.

Although such comparisons are speculative, they are testable by brain scans. The question is whether religious thoughts and their secular counterparts are processed in the same God Spots. An undeveloped area of research in neurotheology is identifying what kinds of secular thoughts are processed in God Spots. If secular and religious thought are processed in the same neural sub-network, we may consider the two kinds of thought are comparable. For example, Schjødt et al. (2009) contend that their study indicates that praying to God is an interpersonal communicative experience comparable to normal interpersonal interaction.

**Table 1.** Religious Thought Categorized in Secular Terms.

| Religious Thought | Secular Counterpart |
|---|---|
| Faith | Belief/trust |
| Hope | Hope/petition |
| Agape Love | Caring/empathy |
| Prayer—adoration | Respect/praise |
| Prayer—petition | Fear/anxiety |
| Prayer—thanksgiving | Gratitude/appreciation |
| Prayer—confession | Shame/guilt |

The brain-scan literature reveals that the processing of secular and religious thought arises from the recruitment of specific networks. Thus, we can postulate that brain scans would identify the topography of their processing networks. Moreover, we might detect the distribution and coordination of processing among several loci. Thus, we could hypothesize that brains create a thought-type catalog that automates associating certain brain areas with the processing of certain kinds of thought, irrespective of their secular or religious context. This hypothesis extends the well-established understanding of topographically selective processing of somatosensory and motor functions. Such a thought-register system could account for the brain's ability to generate a unified representation of the various cognitive components of a stream of coherent thought.

Creating a catalog that associates specific sets of brain areas with specific kinds of cognition, both secular and religious, is no small task for future research. Yet, if and when that task is completed, we are in some sense back where we started: What will we make of such knowledge? Maybe we will find new ways to think about cognitive categories. How will it advance cognitive neuroscience? Will it have any practical application in advancing spirituality ... or anything else (education, medicine, psychiatry, psychology, social science, etc.?). Such questions are not asked nihilistically. Rather, the point is for scholars in all fields to begin thinking more about "Now what?" We might productively address this new "Now what?" moment by exploring avenues of pursuit that might have theoretical or practical value.

For any given category of cognitive activity, both secular and religious, here are a few questions that might be helpful:

- Is the allocation of neural resources to certain areas random? Or are there clear signs that some kind of hard wiring governs which areas are recruited?
- Are the areas recruited the same in every individual? If not, what individual characteristics and past learning experiences might be involved in individual differences?
- Do the recruited areas in a given individual remain the same over time? Or do they differ depending on contexts or emotional state or extent of learning?
- Does intense repetition of a religious cognitive activity lead to the participation of fewer brain areas during such activities, as is commonly observed for secular thinking? If so, what factors most greatly influence the number of engaged brain areas?
- What factors influence the rate at which certain brain areas drop out of engagement as a result of repeating the same cognitive experience?
- As certain brain areas drop out with repetition of the same cognitive activity, what other kinds of cognition can be engaged in the "liberated" brain areas?
- Do the released resources have a bias for processing other kinds of thought?
- Assuming that it is good that repeated experience and learning reduce the neural resources needed, what situations or practices could intensify and accelerate the ability to process thought using fewer resources?

## 2.8. Static and Functional Anatomy

While the emphasis in this present analysis is focused on real-time religious experience, underlying anatomy may determine where and how religious cognition is processed. A structural MRI evaluation of religious people revealed that their presumed intimate relationship with God and engagement in religious behavior was associated with increased volume of the right middle temporal cortex. Fear of God was associated with decreased volume of left precuneus and left orbitofrontal cortex. A cluster of traits related to pragmatism and doubting God's existence was associated with increased volume of the right precuneus. Variability in religiosity of upbringing was not associated with variability in cortical volume of any region (Kapogiannis et al. 2009a). However, another structural MRI study of grey matter volume (van Elk and Snoek 2020) found no structural brain differences. This suggests that studies of religiosity should focus on dynamic functional changes rather than underlying anatomy.

Both studies lacked a longitudinal design that checked for changes in the same individual over time with or without repeated religious experience and commitment. "Born again" individuals might be of particular interest to study in terms of comparing the God Spots before conversion and at various times afterward.

In all kinds of religious and secular thought, activity in a given neural "hot spot" could change its connectivity with brain areas with which it normally coordinates. For example, in the study of people with anxiety by Etkin et al. (2009), the basolateral amygdala was less connected than normal with all of its usual targets (sensory and medial prefrontal cortices) and more connected with targets of the centromedial amygdala (midbrain, thalamus, and cerebellum). Both areas were less connected with brain areas associated with assigning salience to sensation. However, the amygdala had greater connectivity to a fronto-parietal cortical executive-control network previously found to exert cognitive control over emotion. We do not know much about the time scale over which connectivity changes occur in either secular or religious thought.

## 2.9. Recruitment of Processing Circuitry

All cognitive tasks require some sort of resource allocation mechanism so that certain portions of the global neural network are recruited for the task. Some allocation of resources is hard-wired for certain kinds of information input. For example, painful stimuli must be routed to the spinothalamic and thalamo-cortical pathways that are set aside for such stimuli. But cognitive and affective processes have no such obligatory direction, or at least so we think. Almost nothing is known of how the brain "decides" where to send a given kind of cognitive information for processing in its neural networks. Comparing brain scans during religious and secular cognition might shed light on this selectivity and on how rigid it is.

Perhaps we should consider the hypothesis that the brain creates thought and feeling categories and allocates neural processing resources accordingly. This allocation could be topographical, not in the sense of mapping body parts, but rather in terms of mapping segments of the global neural network according to cognitive and affective category.

Secular studies have informed us that certain kinds of sensory input are routed to certain brain areas (olfaction to the limbic system, vision to the occipital cortex, sound to the temporal cortex, somatosensory inputs to the sensory cortex, and so on). Those inputs are then selectively routed to other areas for cognitive processing. Secular brain-scan studies seem to indicate that certain brain areas are biased to process certain kinds of secular cognition that have religious counterparts (Table 2).

Obviously, changing the nature of cognitive thought could recruit different neural circuitry, as has been abundantly documented in brain-scan reports involving secular thought. The various God Spot reports seem to suggest the same phenomena, but few studies have examined shifts in neural processing areas in response to changes in the kind of religious thought in the same subject. The unanswered question remains: What brain areas become activated, and in what sequence, as the nature of thought changes? Additionally, one may consider whether subjects are consistent in generating any specific

religious thought and in which God Spots become activated. If each person has a unique brain-area activation profile, what genetics or life experiences might account for individual differences?

**Table 2.** Brain Areas Associated with Specific Kinds of Secular Thought.

| Kind of Thought | Non-Religious Studies |
| --- | --- |
| Belief/trust | Ventral striatum, medial prefrontal cortex (Fareri et al. 2015) |
| Hope/positive expectation | Prefrontal cortex (PFC), premotor areas (Leaver et al. 2009) |
| Caring/empathy | Ventromedial prefrontal cortex, medial orbitofrontal cortex (Ashar et al. 2017) |
| Fear/anxiety | Amygdala, insula (Etkin and Wager 2007) |
| Gratitude/appreciation | Medial prefrontal cortex (Kini et al. 2016), ventromedial prefrontal cortex (Karns et al. 2017) |
| Shame/guilt | Shame: middle frontal gyrus and parahippocampal gyrus; Guilt: fusiform gyrus, middle temporal gyrus (Michl et al. 2014) |

Specifically, knowing the topography of localized processing of different kinds of religious thought could provide some understanding of how neural resources are allocated and coordinated among network domains in the global workspace as different kinds of thought are consciously attended and processed. Brain-scan studies might also identify the recruitment of brain areas that are engaged in analogous secular and religious thought. Are the same brain areas activated during different kinds of prayer (adoration, thanksgiving, petition, or confession)? Are any differences in God Spots reversible as the person switches back to a prior kind of prayer? The same kind of questions can be asked of secular thought in the same cognitive category as a given kind of religious thought.

Tracking dynamic changes in activated brain areas with changes in intensity of cognition provides an additional way to study neural resource allocation. This can be done for secular and religious thought. For example, experimental subjects could be instructed during fMRI monitoring to begin a prayer expressing love of God in a generic, non-personal way. Then, prayer can shift to loving God for personal help received, or for forgiving one's sins, or for offering eternal life in paradise. As prayer content intensifies, do different brain areas become engaged? Do some areas become less active? Do the shifts change in predictable ways?

Although prayer may be the easiest religious activity to study using brain scans, it also seems feasible to study some other kinds of religious behavior. The motor component of religious practice will surely engage certain brain areas involved in movement in association with cognitive content. One example of religious behavior that seems amenable to this kind of brain scanning would be to compare the Catholic practice of praying the rosary with and without finger movements. Another possibility is to compare God Spots during kneeling, with and without the accompanying prayer.

## 3. Suggested Research Strategies

Pursuit of the various growth areas for neurotheology requires appropriate strategies and tactics. New methods would be desirable, but we are currently constrained by the existing technologies of metabolic scans, which are only proxies for neural signaling and processing, and electroencephalography (EEG), which only directly monitors signaling and processing in the neocortex.

### 3.1. Imaging Techniques

Most brain scan studies are conducted with magnetic resonance (fMRI) or electrical recording (EEG). The fMRI studies have the advantage of localizing activity to particular brain areas, cortical and subcortical. This kind of brain imaging can be faulted on technological, methodological, and philosophical grounds (Shifferman 2015). A major disadvantage is that images are based on small metabolic changes that can be no more than proxies for neural signaling.

The validity of fMRI is called into question, given that thoughts inside a noisy MRI scanner are unlikely the same as normal thoughts. Ladd et al. (2015) suggested that the current fMRI technology may be overly disruptive for cognitive research, in that it is hard to believe that "wearing ear protection inside a 60-cm tube with 110 dB of ambient noise would be similar to typical prayer practice." Prayer-on-demand in any experimental context is odd compared to how people usually pray, and the fMRI context exacerbates that problem. The same problem exists for studying other kinds of spiritual practice.

This limitation is greatly reduced by using EEG methods, which also have the advantage of reflecting actual neural processing on rapid time scales. The EEG is especially appropriate in God-spot research because the electrical signals are proxies of the actual signaling and processing that occur in neural networks. In essence, information about God or anything else is captured in the brain in the form of neuronal impulses that mentally represent, process, convey, and store the information.

Different network zones register sensations, memories, and thought in terms of patterns of electrical pulses (nerve impulses) that propagate and often reverberate therein. We can think of impulses as related to thinking as money is to commerce; that is, impulses are the currency of thought. Impulse representation of information is distributed and processed in selectively recruited sub-networks in the brain's global network. Processing these impulses yields decisions about the relevance and need for an active response, which occurs cognitively or through body movement (Klemm 2011a). Some portion of these representations may be stored as memory for future use. These mental states operate on both secular and religious information.

A network of neurons dynamically generates patterns of activity that represent and process stimuli and retrieved memories (Harang and Bassett 2010). Local networks can transform and transmit activity patterns across synapses to spatially distributed areas and modify the temporal relationships of activity in the various target areas.

EEG voltage waveforms in various putative cortical God Spots have shifting frequencies and temporal relationships that are indicative of recruitment of neural resources and actual neural processing. The disadvantage is that only activity from the neocortex is accessible from scalp electrodes, with subcortical activity monitored only indirectly without spatial resolution. Also, spatial resolution is poor because of the volume conduction of electrical signal. Both fMRI and EEG scanning methods present unavoidable statistical problems in data analysis.

*3.2. Design and Analysis Principles*

With either scanning method, it seems important to emphasize several principles that are key to learning the most about cognitive processes involved in specific kinds of thought, secular or religious. Experiments should achieve cognitive granularity. That is, they should ensure rigorous control over the kinds of thought being imaged. It is not enough, for example, to monitor subjects who have merely been instructed to pray, but instead the kind of prayer should be specified. Granularity of experimental design can be achieved, in several ways:

1. **Focusing on inter-area coordination.** Brain-scan studies of secular cognition make it clear that most kinds of thought engage and coordinate activity in more than one brain area. Indeed, despite the common assumption that speech processing is specific to Broca's and Wernicke's areas of the left cerebral cortex, there is linked activity in other areas, and the speech areas can respond to nonverbal (Price et al. 2005);

2. **Distinguishing proximal and distal causes.** Thought and brain scanning should be tracked in time to permit detection of brain areas that automatically respond to a pre-specified function and those that are then recruited to provide extra neural resources for the needed processing;

3. **Identifying adaptation and learning effects.** Repetition of any kind of thought produces learning effects, which affect the amount of neuronal resources needed for processing. Monitoring how brain area activity changes over a series of studies using the same methodological protocols will shed light on how the brain recruits and allocates its processing resources;

4.  **Identifying the different cognitive processes that activate the same brain areas.** Certain secular and religious thoughts may activate the same brain areas. Experimental designs that map brain activity in the same participant while thinking different kinds of secular and religious thoughts will allow for the creation of a generic cognitive catalog.

*3.3. Interpretation of Data*

Cognitive neuroscientists who use imaging to infer cognitive processes must be careful to avoid the mereological fallacy of ascribing psychological attributes to parts of brain that can only intelligibly be ascribed to a larger system within the brain. The functional network architecture of the brain clearly mandates the conclusion that although a given cognitive function may have a localized concentration of processing, it is neither isolated nor independent of operations in other parts of the global workspace network.

A core interest to cognitive neuroscientists is the mechanisms by which the brain allocates neural resources within different parts of the global network. Allocation seems to require some kind of control system. For example, an analysis of hundreds of fMRI studies (Cole and Schneider 2007) revealed high inter-regional activity correlations of six coactive cortical areas regions during rest and task performance, which seem to form a functionally connected cognitive control network: anterior cingulate cortex/pre-supplementary motor area (ACC/pSMA), dorsolateral prefrontal cortex (DLPFC), inferior frontal junction (IFJ), anterior insular cortex (AIC), dorsal pre-motor cortex (dPMC), and posterior parietal cortex (PPC).

Mapping studies involving secular and religious categories of thought need to include these six areas as regions of interest. Areas that they recruit could be identified by those areas that become more active for a specific cognitive task. Analysis of the time lags between activation of areas might elucidate how the brain allocates processing resources. In the case of EEG monitoring, similar synchronous activity can be identified by testing for frequency-specific coherence of voltage waveforms in various scalp electrode sites.

We do not know whether the allocation of neural resources is an unsupervised, quasi-random process in biological neural networks. We have no information on whether resource allocation process is the same for secular and religious cognition.

Although the human neocortex has an executive network, we do not know the full extent of that function. Moreover, casual observation of human behavior makes it clear that many attitudes are not well-controlled through supervisory executive action.

## 4. Conclusions

God Spot research thus far has provided a foundation for moving beyond phenomenology to robust hypothesis generation and testing of the cognitive processes, shared and unshared, for various kinds of secular and religious thinking. We are now positioned to refine and exploit the findings of brain scans as they relate to religious experience. In particular, we should exploit the cognitive neuroscience that is common to secular and religious thought, within a framework that includes: (1) neural network operations, (2) cognitive content of prayer, (3) biology of belief, (4) measures of religiosity, (5) role of the self, (6) learning and memory, (7) religious and secular cognitive commonalities, (8) static and functional anatomy, and (9) recruitment of neural processing circuitry.

Science is inevitably central to understanding religiosity. In the present era, neuroscience provides great relevance for cognitive biology. How well this point applies to theological thinking has thus far been rather uninteresting. Perhaps neuroscience will one day help us explain some aspects of religiosity if the field of neurotheology expands in the directions suggested in this analysis.

**Funding:** This research received no external funding.

**Acknowledgments:** The author appreciates the editing and suggestions provided by colleagues Ava English and Barbara Gastel.

**Conflicts of Interest:** The author declares no conflict of interest.

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
