# Peer review of "God Spots in the Brain: Nine Categories of Unasked, Unanswered Questions"

_religions, doi:10.3390/rel11090468_

Round 1
Reviewer 1 Report
This paper provides a fascinating, concise review of extant, relevant research and proffers valuable suggestions for future research in this developing area of endeavour.
It is very readable and not too heavy on jargon, although a basic understanding of brain physiology would be advantageous for any reader.
However, this whole paper is let down by line 478. You cannot just use a throw-away line without justification or recognition of the limitations involved. Science (as a limited way of knowing) is inevitably (really?!) central to an understanding of religiosity (which is multi-faceted and inadequately defined in this paper).
Line 17 is much less pedantic and more promising.
Typos line 390, '.' not '?' after 'prayer';
line 449 delete 'of' after 'mechanisms'.
Author Response
The reviewer's supporting and helpful comments are appreciated. I rewrote the last line and made the deletion of "of" in line 449.
Reviewer 2 Report
Strengths:
1. This paper does well to argue that if neurotheology is to be of interest to scholars in other disciplines (like religious studies and theology) then it needs to move beyond merely identifying ‘god spots’. The author is spot on when she/he writes that many theologians and religious studies researchers are left with the “so what?” question, when presented with much neurotheology.
2. The author shows good knowledge of the neuroscience and the recent literature in this ever changing field – even if, at a few points, this is hard to follow for the non-specialist. As such, this paper provides an excellent overview of the current state of neurotheology.
3. Many of the prices of advice given to the field of neurotheology appear to me to be sound. The need for neurotheology to move from “spots” to “networks” is a good employment of this knowledge to the area of neurotheology. The need for more focused prayer studies, examining different types of prayer, seems sound. I agree that these would benefit the field.
Weakness:
1. Some minor comments:
a. P.1, line 38-30: I don’t think different denominations can be descrbied as “scholarly hybrids” or seen as equivalent to different sub-disciplines within neuroscience. An easier comparison would be the sub-disciplines within theology (i.e., sacramental theology, practical theology, philosophical theology, historical theology, ecotheology, etc. which combine theology with another academic year, like social anthropology, or philosophy).
b. P.1 line 41: I’m unsure what is “triune” in this sentence.
2. Although the advice given to neurotheology seems sound (see, pt. 3) – the author delivers this advice very briefly (in a sentence or two), after surveying the current literature. Since these pieces of advice are the main take-away point of the paper, these each need to be more thoroughly developed. The author should argue for how following her/his advice might change the field, perhaps even offer a tentative hypothesis or two to be tested, with some reasoning for the prediction. In particular, how will following this advice make neurotheology more relevant to those in either neuroscience or theology – since the disinterest of these two parent disciplines was the problem that the author opened this paper with and seeking to help the field of neurotheology move beyond. For example, why think that might more focused prayer studies be any more interesting to religious studies/theology scholars than general prayer studies?
3. On page 4, line 180-182 the author asks: “When brain areas X, Y, Z, become active during a religious experience, has some neural process opened the gates to a neural network that allow God to enter within us?” The author implies that this is the sort of question that neurotheology might seek to answer? How? Is this really a question that brain scans and neuroscience can answer? Or is it a theological question, that needs a different kind of methodology (as found in theology itself)? In short, whilst the author discusses the question of new strategies and methodologies, the author does not challenge the standard neurotheology method of using the tools of neuroscience to explore spirituality, religious phenomena and theology. Might it be this very methodology that limits neurotheology’s findings in the way that the author wants neurotheology to move beyond.
4. P.4, line 188. The author writes, “Belief or disbelief is a decision-making process”. The doxastic status of belief and belief-formation is very controversial, and it is rare to find philosophers to would agree that a person can simply decide what to believe. This section, and the question of doxastic and non-doxastic belief formation could be expand to include such consideration from other disciplines that have paid close attention to this question.
5. P.4-5 On the question of “measures of religiousity,” more need to be said on the possibility of a brain scans revealing that a person is not as religious or fervent in their belief as they claim to be. How is it possible for a person to be deceived about this, and what are the implications of such self-deception? Could this be compared to other work on self-deception? The author needs to do more to motivate the idea that (line 198) “people;s brain are less able to lie.” Whilst this appears simple, the role of interpretation in neuroscience and the individual differences in neurology, suggest that first-person testimony remains more trust worthy on the matter of mental states (beliefs) than fMRIs.
6. I am overall convinced by the authors argument, but the final sentence is far to bold and unfounded by the argument. The author has shown that there are many (9 at least) areas and questions yet to be explored in neurotheology. But she/he has not shown that any of these areas will explain anything or that neuroscience holds any explanatory power for religiosity. Indeed, it seems to me to be one of the strengths of this paper that the author acknowledges that locating God spots and seeing that the brain “lights up” is fairly uninteresting to many scholars outside of neurotheology and doesn’t itself yet explain anything. So, this final sentence is confusing and should either be deleted or made more modest and future dependent, i.e. perhaps neuroscience will one day help us explain some aspect of religiosity if the field of neurotheology expands in the direction the author suggests.
Author Response
Reply to reviewer 2
I wish to thank this reviewer for very useful insight. The manuscript should now be much improved. The six main points have been addressed as follows:
- I revised the comments in line 38 to eliminate “denominations” and replace with the subdisciplines the reviewer mentioned. I revised line 41 to delete “triune.”
- The reviewer very insightfully suggested that I provide better defense for why neuroscientists and theologians should be interested in extended God-spot research. I appreciate the suggestions. I agree that the main thesis needs expansion. I have added comment (new lines 119-133) to respond to the reviewer suggestions. I also moved a couple of paragraphs that immediately preceded this section to a more appropriate location in the text.
- I have revised the text that the reviewer challenged in the Biology of Belief section in the new lines 181-185. I agree with the reviewer’s position in this matter.
- I was unfamiliar with doxastic philosophy, and I appreciate the opportunity provided by the reviewer comments to qualify my original comments. I have now added what I hope is appropriate text in new lines 191-201.
- The point about brain scans and self-deception (lie detection) is well taken. I have modified the text accordingly in new lines 208-214,
- I have modified the last line to be less bold and include the reviewer’s point.
Round 2
Reviewer 2 Report
I am satisfied that the author has revisied the paper and strengthened its arguments. I think this is a value contribution to the current research on neurotheology, and should be published.